# The Effects of White versus Coloured Light in Waiting Rooms on People's Emotions

**Zhihui Zhang \*, Josep Maria Fort Mir and Lluis Gimenez Mateu**

Escola Tècnica Superior d'Arquitectura de Barcelona, Universitat Politècnica de Catalunya, 08028 Barcelona, Spain
\* Correspondence: zhihui.zhang@upc.edu

**Abstract:** Lighting ambience in architecture is one of the important factors affecting the emotions of people, and the study of the psychological needs of architectural lighting may provide more rational guidelines for architectural design. There are many previous studies on the emotional impact of lighting in architecture, but most of them use a dimensional model of emotion to analyse emotions, which is difficult for the reader to understand. In this study, we used the dimensional model of emotion to analyse emotions and converted it into easily understood basic emotions through the PAD model. Participants (n = 32) were divided into three groups and subjected to three scenes with different colour combinations. The analysis showed that the arousal and dominance of the participants were significantly affected from white to coloured light. No effect on comfort was observed between white and coloured light. Our study suggests that the use of coloured lighting instead of white lights in a non-clinic windowless waiting room may not improve negative mood.

**Keywords:** colour light; lighting ambience; waiting room; indoor environment; people's emotion

## 1. Introduction

Light affects people psychologically and physiologically, and its effects have been studied in a variety of contexts, such as the physiotherapeutic effects of light, the effects of light on task performance and job satisfaction, learning efficiency, purchasing behaviour and comfort [1–3]. Lighting plays a key role in human life, especially as a central tool in architecture and interior design. With the advancement of LED technology, lighting systems not only control illumination and correlated colour temperature (CCT), but also adjust chromaticity. Thus, based on LED lighting systems, we can design more attractively coloured ambient light. For example, modern aircraft cabins use coloured light to influence the thermal sensations of passengers [4], and coloured lighting in car interiors improves satisfaction and comfort [5]. The emotional impact of coloured ambient lighting on people, especially in indoor environments with the intention of improving mood, deserves further investigation. Studies have shown that some people have negative experiences in different types of waiting rooms [6–8]; perhaps we can improve the mood of people in waiting rooms with the help of coloured ambient light.

In this study, we investigate whether positive emotional effects could be induced in people with coloured ambient light in waiting rooms. The argument of this paper is developed as follows. We first critically review the current state of knowledge in the field that defines the measurement of emotional responses with respect to light colours, specifically considering the relationship between white lights, coloured lights, and emotions. We found that most of the previous studies on the effects of light on human emotions used dimensional models of emotions for emotion analysis. With dimensional models of emotion, differences and similarities between emotions can be easily estimated, and the vector of values used to represent them is continuous and well suited for analysis. However, the core emotions of the dimensional emotion model are difficult for the architect to

understand and are not better explained relative to the discrete model emotion [9]. To improve this gap, we used the PAD (pleasure, arousal, and dominance) model [10] to classify dimensional emotions into discrete emotions in order to obtain the basic emotions. We examine the emotions of participants with respect to white and coloured lights through traditional observation and questionnaires. The hypothesis is that applying coloured ambient lighting in architectural spaces may elicit positive human emotional responses. We conclude with a discussion of some of the limitations of this study and directions worth pursuing in future research phases. Therefore, we expect that this study will help some architects provide more rational reasoning for their choice of lighting and help some designers understand the emotional impact of users in indoor spaces.

## 2. Literature Review

### 2.1. Waiting Rooms

Creating a vibrant waiting room in which people can sit or stand until the event or appointment for which they are waiting begins is perhaps one of the underlying goals of good architectural design. However, some designers often ignore waiting spaces, so most people rate the waiting rooms of clinics as unattractive spaces [11]. Studies have found that clients spend an average of 55.27 to 71.28 min in patient waiting rooms, approximately four times as long as receiving the actual treatment [12,13]. Similarly, studies in consumer, office and display categories have revealed significantly adverse effects on individuals, leading to customer dissatisfaction [6–8]. It is difficult for architects to reduce customer waiting times through design optimisation, but it may be possible to make customers more comfortable while waiting by improving the environment of the waiting room. Positive emotional guidance might help to reduce the mental and physical stress of clients. By carefully handling lighting, sound and view, it is probably feasible to create a welcoming atmosphere in the waiting room. By doing so, the comfort level of patients and visitors might be increased. For example, setting up quiet areas decreases the decibel level of conversations, and adding natural ambient sounds or music can reduce anxiety [14–16].

Likewise, visual art might positively impact a client's waiting experience [17] and adding greenery, whether artificial or natural, can increase customer satisfaction and reduce negative emotions [7,17,18]. However, studies focused on the emotional change between white and coloured lighting remain scarce. Some architects believe that colourful lighting design helps create an attractive space. Coloured light may be important as an emotional evocation factor to enhance the attractiveness of a waiting room. Bilgili et al. experimentally concluded that the colour of the lighting determines the service quality perceptions and service quality evaluations of customers. Together with their eventual future repeat purchase intention, customers perceived that green light makes the waiting period feel relatively shorter [19].

### 2.2. Emotional Response to White Light

Most studies on the emotional impact of white light have focused on two parameters: illuminance and correlated colour temperature. A number of laboratory-based studies have revealed that changes in white light illumination values (below 5000 lx) have a limited effect on people's moods. Only very high illuminance (5000–10,000 lx) could create positive emotions [20–22] (Table 1 part ①). For instance, Goel et al. showed that exposure to high illuminance light (10,000 lx) for 15–30 min might significantly reduce depression and anger [20]. In a similar way, Leichtfried et al. suggested that high illuminance (5000 lx) positively affects mood, and they did not find high illuminance to cause a negative attitude [22]. In a study conducted in Austria, Hoffmann et al. asserted that variable illuminance values (500–1800 lx) have a more noticeable effect on subjective mood and perception than a fixed illuminance value (400 lx) [21]. Other studies found no significant effect of illumination on human mood.

CCT has been shown to be more effective than illuminance in influencing people's emotions (Table 1 part ②). Hidayetoglu et al. found higher scores for attractiveness and memory for warm than for cold CCT and found higher median scores for positive perceptions with warm CCT [23]. Li et al. suggested that under high luminance LED lighting, participants showed higher positive emotions at 4000 K than at 6500 K [24]; similarly, a study by Hutchings concluded that warm light is more comfortable than cool light [25]. Hsieh concluded that 2700 K is more conducive to evoking positive emotions than 6500 K [26]; however, CCT has little effect on mood in general lighting. A different finding by Iwata indicated that 5200 K light has a more comfortable and relaxing impact on participants, compared to 2800 K light [27].

**Table 1.** Studies on the impact of white light on emotion.

| Factors | Year | First Author | Sample Size | Exposure Duration | Experiment Conditions | Emotion Effect | Emotion Measurement Method |
|---|---|---|---|---|---|---|---|
| Part① illuminance of light | 2006 | Goel [20] | 69 F + 49 M Age = 19.4 ± 1.7 | 15–30 min | 10,000 lx | N↓ | Q |
| | 2008 | Hoffmann [21] | 11 M Mean age = 25 Age range 22–34 | 3 days | 500–1800 lx; 400 lx | Variable light P↑ | Q Urinary test |
| | 2015 | Leichtfried [22] | 17 F + 16 M Age = 33.0 ± 7.2 | 30 min | 400 lx; 5000 lx | P↑ | Q |
| Part② Correlate colour temperature of light | 2012 | Iwata [27] | 12 F + 7 M Mean age = 21.7 | 240 min | 2800 K; 5200 K | 5200 K P↑ | Q |
| | 2012 | Hidayetoglu [23] | 60 M + 60 F Age range 19–25 | 25 min | 2700 K; 4000 K; 5300 K | 2700 K P↑ | Q |
| | 2015 | Hsieh [26] | 24 F + 39 M Age range = 18–28 | 60 min | 2700 K; 3000 K; 4000 K 6500 K | 2700 K P↑ | Q |
| | 2017 | Li [24] | 6 F + 14 M Age = 24.7 ± 1.5 | 13 min | 4000 K; 5000 K; 6500 K | 4000 K P↑ | Q |
| | 2019 | Burattini [28] | 20 F + 20 M Age = 22.6 ± 1.35 | 15 min | 3000 K; 6800 K | 6800 K P↑ | Q |
| | 2019 | Li [25] | 16 F + 14 M Age = 25.4 ± 2.9 | 60 min | 40.70 cd/m². 18.36 chroma. Hue range 0–270°; 0–135°; 135–270°. | 2700 K P↑ | Q |

Abbreviations: female (F); male (M); increased positive emotion (P↑); decreased negative emotion (N↓); questionnaire (Q).

*2.3. Emotion and Response to Colour Light*

We can render ambient light using coloured light, or through diffuse reflection or the transmission of colour from walls, glass and other objects. There is an abundance of research on interior wall colour and the mood of participants, and most studies have found a correlation between emotion and colour [23,29–34]. For instance, Kwallek et al. suggested that light blue–green offices are more pleasant to work in than offices painted in other colours [34]. Similarly, a study on college dormitories found that blue environments produce a calming mood and promote learning activities [32]. Lipson-Smith et al. described that a blue coloured room makes people feel calm [33].

However, studies on the effect of coloured ambient lighting on human emotions are scarce. The results of these studies are not consistent, with most concluding that coloured light has a positive effect on people's moods, but some studies have found that coloured light has a negative effect on people's emotions (Table 2) [35–38]. Studies such as Plitnick's found that red and blue light increases EEG beta power (12–30 Hz), reduces drowsiness, and increases positive emotional effects [36]. In the study of Raymann, the combination of light colours with blue components had a perceived response that affected comfort and safety [37]. However, Wilms' study showed a significant increase in the participants' heart

rate in coloured light [35]. Nevertheless, we argue that these few studies do not provide a clear picture of the effect of coloured ambient lighting on mood.

**Table 2.** Studies on the impact of colour light on mood.

| Factors | Year | First Author | Sample Size | Exposure Duration | Experiment Conditions | Emotion Effect | Emotion Measurement Method |
|---------|------|--------------|-------------|-------------------|------------------------|----------------|----------------------------|
| Colour of light | 2010 | Plitnick, B. [36] | 24, 19–27 | 240 min | Red and blue light | P↑ N↓ | Q EEG ECG |
| | 2011 | Raymann, Roy [37] | 18 F + 19 M Age = 21.3 ± 2.6 | 68 min | Red–green, red–blue, green–blue, and RGB | Blue P↑ | Q |
| | 2018 | Wilms, Lisa [35] | 49 F + 13 M Age = 23.37 + 6 | 15 min | Blue, green, red, and grey | Red N↑ | Q |
| | 2014 | Wang [38] | 10 F + 10 M Age = 23.4 + 2.3 | 60 min | 72 hue colours | P↑ | Q |

Abbreviations: female (F); male (M); increased positive emotion (P↑); decreased negative emotion (N↓); increased negative emotion (N↑); questionnaire (Q); electroencephalogram (EEG); electrocardiogram (ECG).

*2.4. Current Gaps in Knowledge and the Potential Contribution of the Proposed Approach*

There have been many previous studies on the effects of colour on mood. However, most experiments have used computer screens, virtual environments, colour blocks, or text to present colours rather than conducting experiments in a real environment. Participants need colour imagery, which can affect the outcome of the experiment. Furthermore, most previous studies have not used white light as a reference baseline. For example, in Plitnick's study, his reference measurement was made in a dim light environment [36]. For Varkevisser, the reference baseline for his experiments was in 20 lx of white light [37]. Most importantly, most analyses in previous studies have used dimensional models of emotion, which are suitable for analysis, but the results of the output are more difficult to understand. Learning from previous studies, our study used LED light sources, with white light as the reference baseline, and a method of transforming into a discrete model of emotion after analysis using a dimensional model of emotion.

Although the assessment of emotions is a complex study, we believe that a combination of qualitative and quantitative research may lead to satisfactory results; it is one of the ways that data can be integrated and shaped in practice. The results of such a study are precious in healthcare, office, and commercial design, where the design impact of waiting spaces is critical. In this study, coloured light does not meet the general indoor lighting requirements of EN 12464-1 [39], but the psychological impact on users of coloured light instead of white light in waiting rooms is worth considering from a scientific point of view. In addition, this type of research may allow architects and entrepreneurs alike to argue for better adaptation of their design solutions to meet human needs. A better understanding of the emotional response to architectural spaces helps some architects to design better and users to make better use of buildings. The interplay between architectural design and human psychology is essential [40]. Therefore, empirical data on human emotional responses to spatial attributes may be essential for designing successful applications in various domains.

## 3. Methodology

### 3.1. Experimental Setting

Our main objective was to measure the relative impact of white light and coloured light on people's emotions in waiting rooms. For this purpose, three experimental scenarios were designed: Scene 1 (white light to blue–green light), Scene 2 (white light to green–red light), and Scene 3 (white light to blue–red light); (Figure 1).

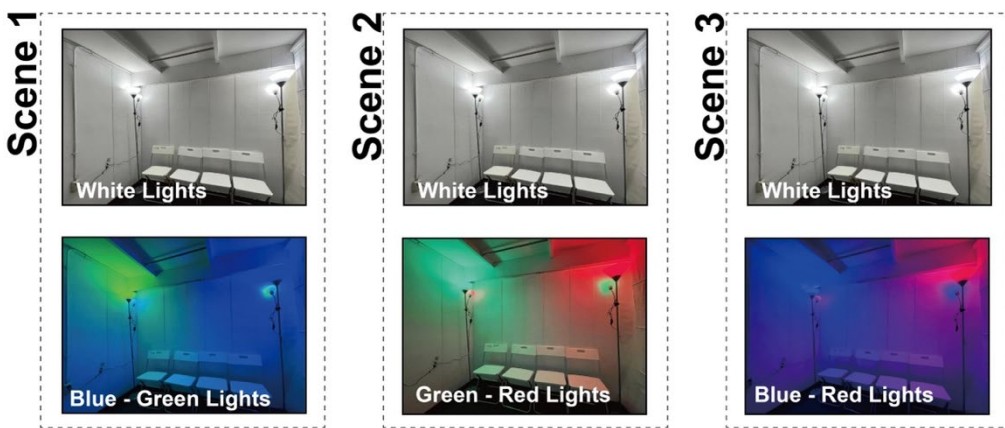

**Figure 1.** Experimental ambient light combinations.

The study was performed in an experimental environment, and the LED light scenes were conducted in the laboratory space. The space was arranged as a four-seat waiting space with dimensions of 3.8 m × 2.6 m × 2.3 m. The walls and ceiling were painted white, and the floor was dark grey. All the windows were covered with white KT panels to reduce the impact of factors besides the light source of the experimental setup, such as views, vegetation, natural light, furniture or smell and sound. The room temperature was set at 23.5 °C, and the humidity was 56%. The furniture in the room was arranged as follows: Four chairs, with two luminaires on either end of the row of chairs. In order to make the colours blend more evenly, we chose a luminaire with two light sources, one main light source upwards and one small light source at an angle of 60 degrees upwards from the horizontal. According to the CIE classification, the luminaire selected was a semi-indirect luminaire due to the direct luminous flux to the working plane between 10% and 40% [41]. For white light, we used LED 1 as the main light source of the luminaire and LED 2 as the small light source of the luminaire; for coloured light, we used LED 3 as the main light source and LED 4 as the small light source (for light source type, see Table 3). The colour of the lights was controlled using the Muvit IO app, using the software's default colour settings for red, blue, and green lights. We alternated the two-colour light sources on each luminaire. For example, if the main light source of one luminaire was red and the small light source was blue, the main light source of the other luminaire would be blue, and the small light source would be red. We measured the relative spectral distribution of the four ambient lights with a Sekonic C700R spectrometer, as shown in Figure 2.

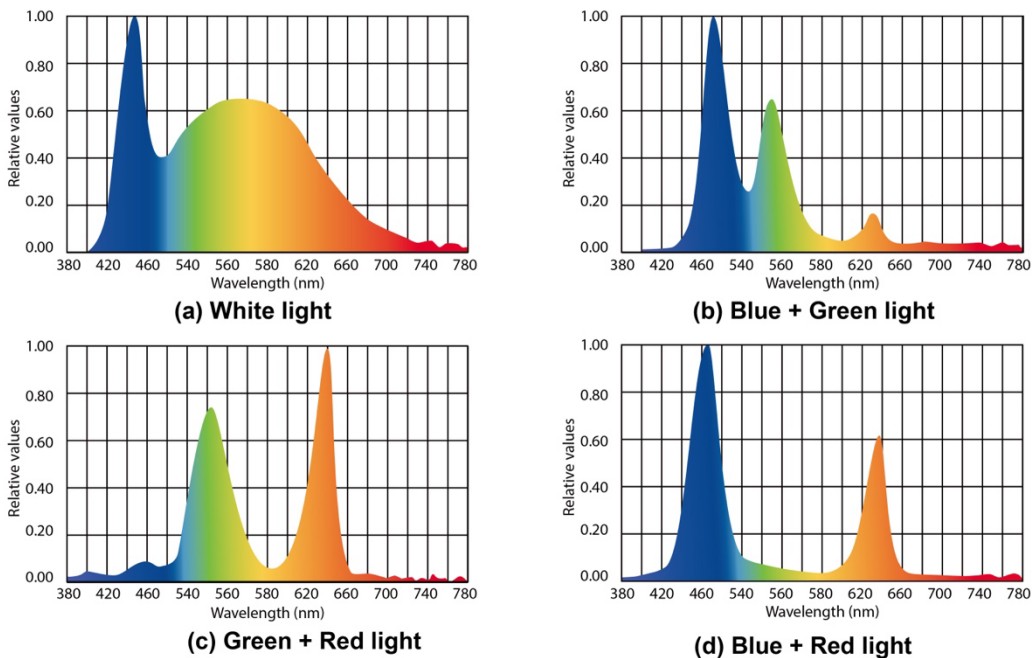

**Figure 2.** Relative spectral distribution of the four ambient lights.

**Table 3.** Selected parameters of the light sources.

| Light Source Symbol | P [W] | Φ [lm] | Light Colour | Lamp Base | Manufacturer |
|---|---|---|---|---|---|
| LED 1 | 10 | 1050 | White (CCT = 6500 K, CRI > 90) | E27 | Osram |
| LED 2 | 5 | 470 | White (CCT = 6500 K, CRI > 90) | E14 | Osram |
| LED 3 | 10 | 950 | Colour-Tunable (LED RGB) | E27 | Muvit IO |
| LED 4 | 5 | 470 | Colour-Tunable (LED RGB) | E14 | Muvit IO |

Abbreviations: correlated colour temperature (CCT); colour rendering index (CRI).

### 3.2. Participants

The study was approved by the Ethics Committee of the Polytechnic University of Catalonia, and all participants provided written informed consent. The study participants included thirty-five volunteers recruited in Barcelona. The participants were recruited through local advertising seeking residents living in Barcelona. We effectively recorded the emotions of 32 out of 35 participants (15 females and 17 males), with a mean age of 38.38 years, SD = 14.23; (Table 4). All participants had normal or corrected-to-normal vision (wearing eyeglasses or contact lenses). All volunteers passed the Ishihara colour blindness test. None of the participants had a visual disability. The participants who completed the experiment received a coupon valued at EUR 20. The investigation was conducted in 15 days, following the principles of the Declaration of Helsinki.

**Table 4.** Participant information sheet.

|  | All Groups | Group 1 | Group 2 | Group 3 |
|---|---|---|---|---|
| Male | 17 | 4 | 6 | 7 |
| Female | 15 | 7 | 4 | 4 |
| Age 18–30 | 11 | 6 | 2 | 3 |
| Age 31–40 | 5 | 1 | 3 | 1 |
| Age 41–50 | 8 | 4 | 2 | 2 |
| Age 51–60 | 7 | 0 | 3 | 4 |
| Age over 60 | 1 | 0 | 0 | 1 |

### 3.3. Mood Measurements Questionnaire

The self-reported questionnaire used the Self-Assessment Manikin (SAM) scale, which visually represents Mehrabian and Russell's three dimensions of PAD/VAD (pleasure, arousal, and dominance; Figure 3) Model [10]. The PAD model is a three-dimensional version of the Circumplex model, one of the dominant models of emotional dimensionality theory [42]. The SAM scale uses images to describe the dimensions of each PAD/VAD and provides a nine-point visual scale. For rating pleasure, the SAM ranges from happy (smiling picture) to unhappy (frowning picture). Rating arousal ranges from excitement (eyes open) to sleepiness (eyes closed). For rating dominance, the SAM ranges from tiny to large images, representing uncontrolled feelings to self-controlled or powerful feelings, respectively. The SAM scale has the advantage of being quick to fill in, reducing the self-reported occupancy time, and being more visual and easier to understand, using an image format. The SAM scale was used to record the bipolar dimensions of emotional valence and arousal. Valence (pleasure) summarises how well a person is doing, while arousal (activation) refers to a feeling of mobilisation or energy. The reason we chose the SAM questionnaire is that with the PAD model, we can convert the dimensional model of emotion into a discrete model of emotion with six basic emotions, making it easier for the reader to understand the participants' mood changes. Furthermore, in this experiment, we added a question to rate the participants' comfort.

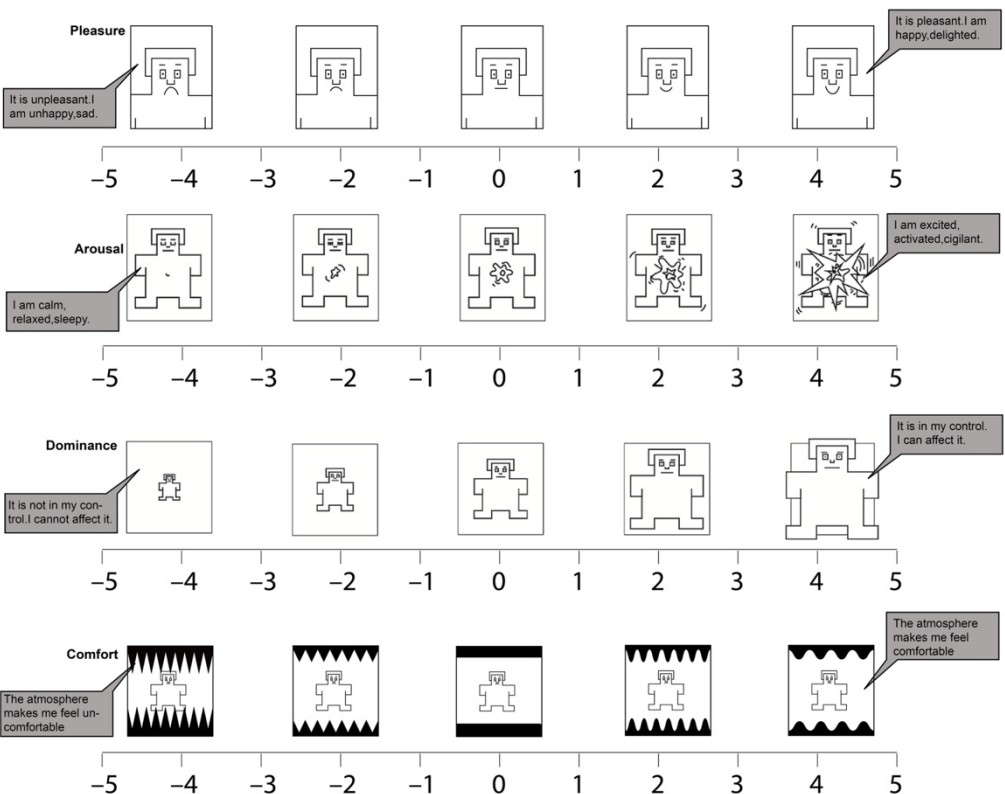

**Figure 3.** The Self-Assessment Manikin (SAM) and comfort measurement scales (pleasure, arousal, dominance, comfort).

### 3.4. Experimental Procedures

Th participants were divided into three groups and entered three set scenarios: Scene 1 (15 min in white light, 15 min in blue–green light); Scene 2 (15 min in colour white light, 15 min in green–red light); and Scene 3 (15 min in white light, 15 min in red–blue light). We balanced the male and female participants in each group and randomly placed them into the different waiting rooms. The participants were informed about the entire experiment, the setup, and the questionnaire scales. After signing an informed consent form, the

participants underwent a two-minute dark light adaptation before the start of the experiment. After light acclimatisation, the participants entered a white light environment for 15 min. Then, the participants filled in their subjective emotions at that moment, according to the SAM scale. After completing the form, the light was modulated to a dark environment, and the participants rested for 2 min before we changed the colour of the light. The participants were subjected to the coloured light for another 15 min, after which they filled in a second SAM measurement (Figure 4). Once the procedure began, participants were not allowed to leave the waiting room, and were required to remain seated until the end of the experiment. The participants were not allowed to use their mobile phones or read, in order to collect accurate results. All tests were scheduled to be performed in the morning in order to reduce errors due to the time of day.

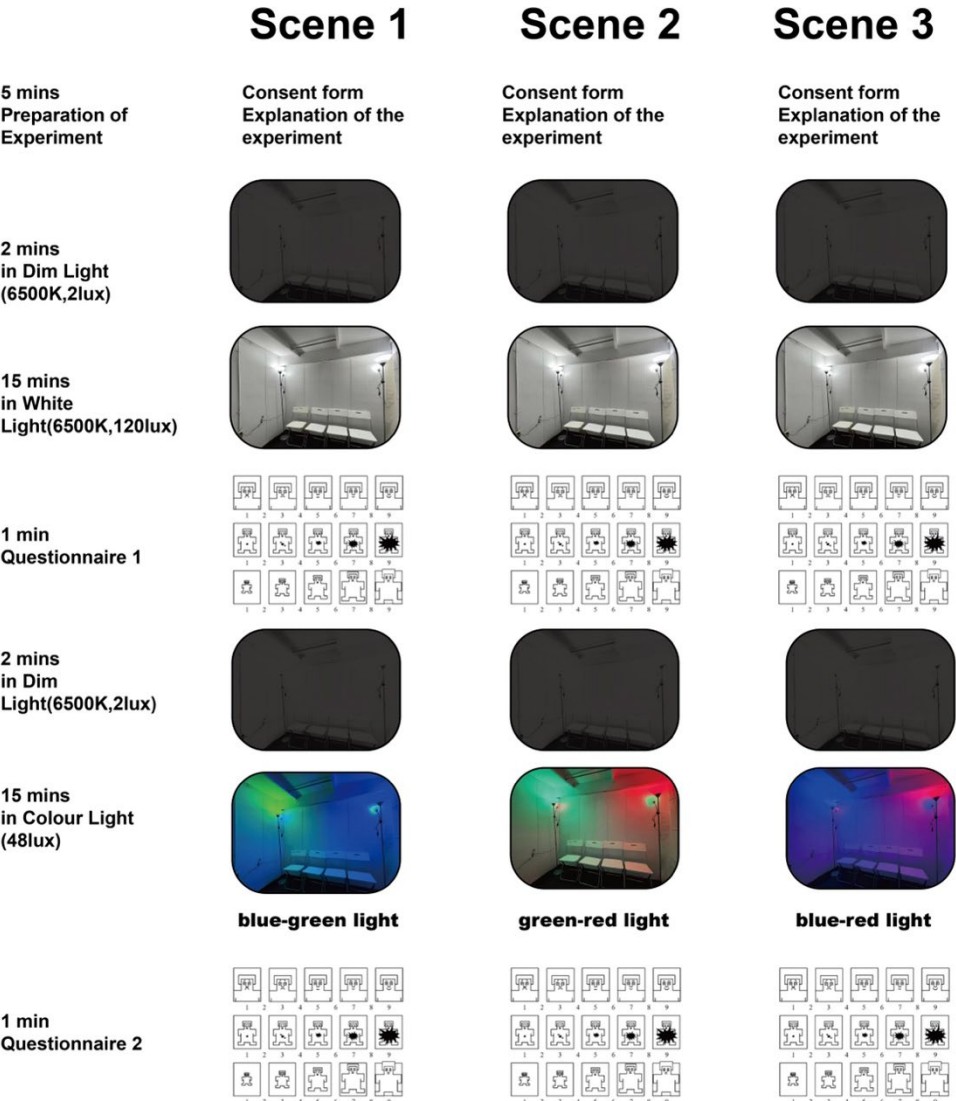

**Figure 4.** Experimental procedure.

### 3.5. Statistical Analysis

Analyses were conducted using the R Statistical language (version 4.1.3; R Core Team, 2022) on Windows 10 × 64 (build 19042). The following steps were followed for each indicator reflecting the degree of emotion. First, the four types of raw emotion data (pleasure, arousal, dominance, comfort) were analysed, and a Shapiro–Wilk (sample size less than 50 [43]) analysis was performed to verify whether the emotion data matched a normal distribution or not. If the Shapiro–Wilk result is $p < 0.05$, then the data are non-

normal distribution data; otherwise, they are normally distributed data [44]. Second, the test method was chosen according to the type of data, using the paired t-test when both sets of data (i.e., white and coloured light) were normally distributed. When one or both sets of data showed a non-normal distribution, the Wilcoxon rank-sum test was used [45]. Third, based on the results of the t-test or Wilcoxon test, we determined whether the differences between the groups were statistically significant or not (indicated by $p < 0.05$). The analysis of the four types of emotion data is independent and can directly observe a significant variation in each emotion data. The filtered data were classified by Mehrabian and Russell's PAD/VAD model to obtain six basic emotions. As long as the change in one of the filtered data is significant, it means that the six basic emotion values converted using the PAD/VAD model might have research significance.

If one of the emotional data is significant, then the statistically significant grouping was selected for classification using the k-nearest neighbours algorithm (*k*NN) method [46], according to the PAD/VAD model of Mehrabian and Russell. Ekman's six base emotions [47] were obtained by classification (Figure 5). By accessing the six base emotions in white and coloured light, we can compare their emotion changes to obtain the relative emotion values.

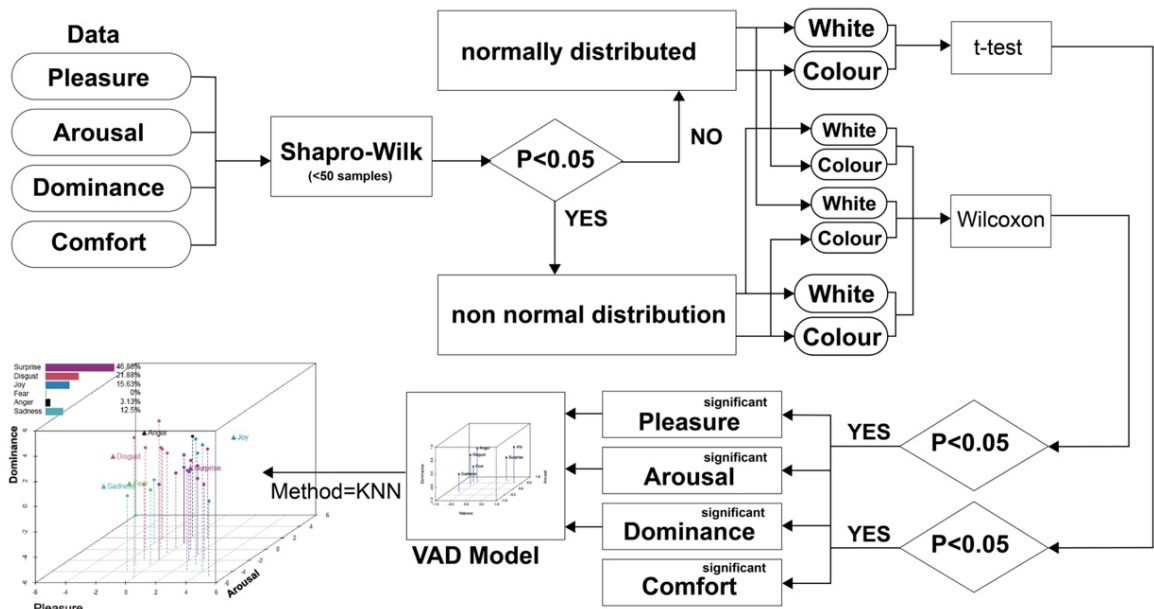

**Figure 5.** Data analysis workflow.

## 4. Results

### 4.1. Rating of Pleasure

Analysis of the data indicated that the pleasure ratings from white to coloured light (all three groups) showed negative but statistically insignificant differences ($p = 0.251$) and small effect sizes (Cohen's d = 0.28). There were no statistically significant differences in Group 1 (white to green–blue light); Group 2 (white to red–green light); and Group 3 (white to red–blue light). Group 2 and Group 3 had moderate effect sizes (Wilcoxon effect size = 0.436; Cohen's d = 0.58,). The significance and effect size results for all grouping analyses showed little variation in pleasure ratings (Figure 6).

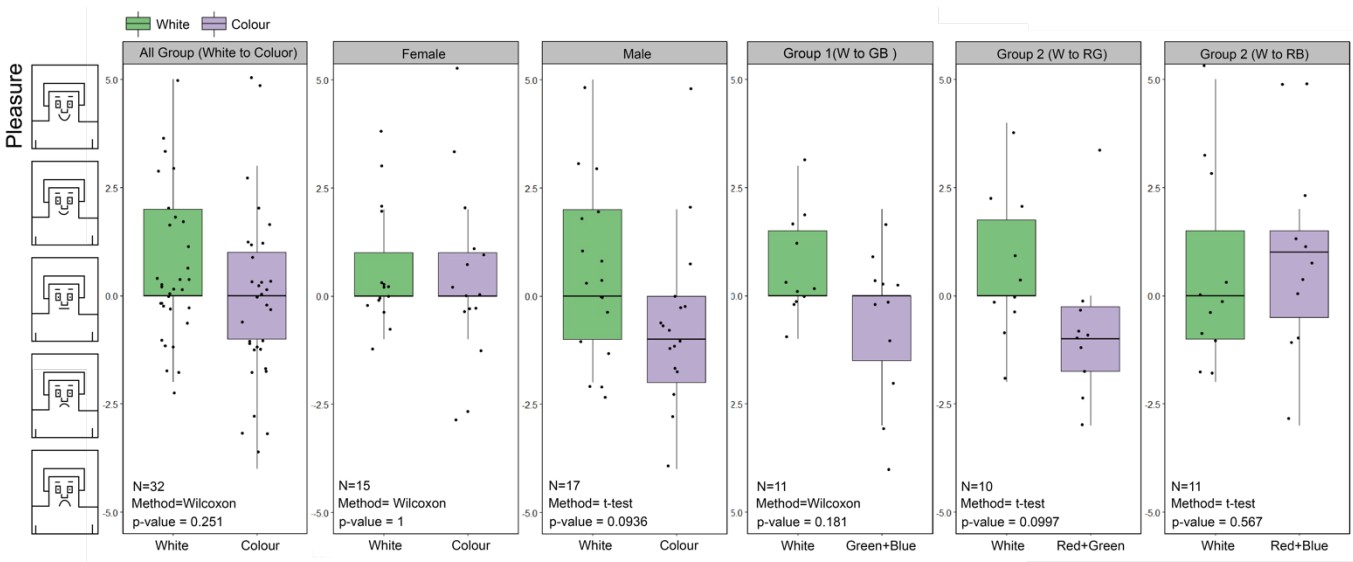

**Figure 6.** Boxplots of valence ratings from white to coloured light (5 = highly pleasant; −5 = highly unpleasant). W to GB, white to green–blue light; W to RG, white to red–green light; W to RB, white to red–blue light; N, number of people. The statistical significance of repeated t-test or Wilcoxon test is marked on top of each measure.

### 4.2. Arousal Rating

The arousal ratings for white to coloured light (all three groups) showed increases with statistically significant differences (*p* = 0.0213) and a moderate effect size (Wilcoxon effect size = 0.412). No statistically significant differences were found in Group 1 (white to green–blue light); Group 2 (white to red–green light); and Group 3 (white to red–blue light). Group 3 had a large effect size (Wilcoxon effect size = 0.633). The significance and effect size results for the other grouping analyses showed little variation in arousal ratings (Figure 7).

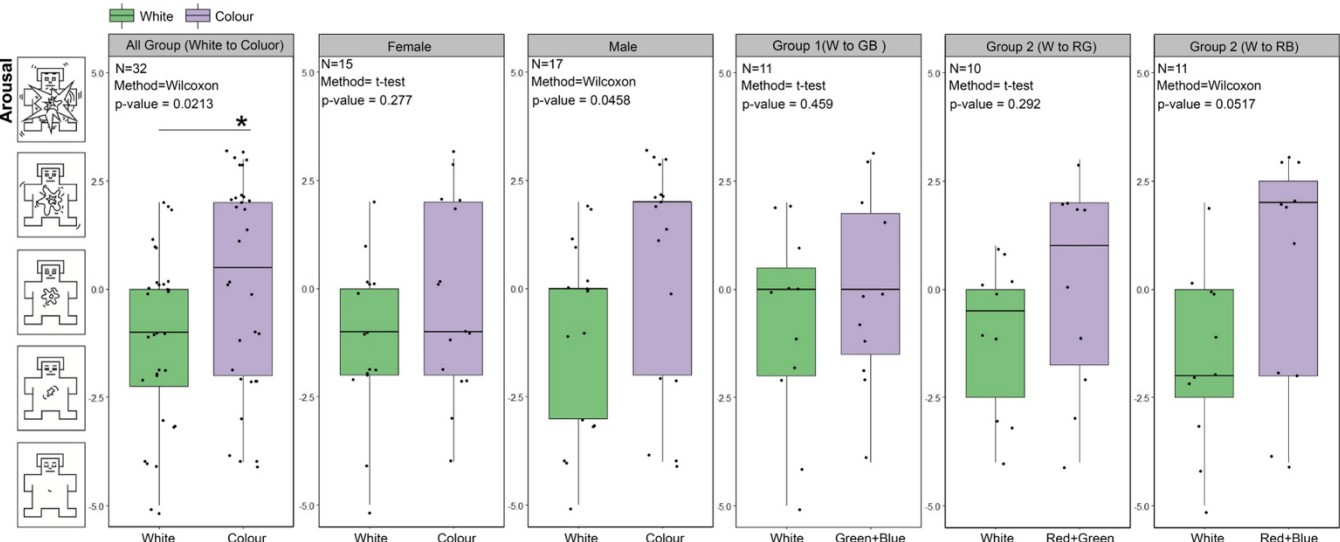

**Figure 7.** Boxplots of arousal ratings from white to coloured light (5 = very aroused; −5 = not at all aroused). W to GB, white to green–blue light; W to RG, white to red–green light; W to RB, white to red–blue light; N, number of people. The statistical significance of repeated t-test or Wilcoxon test is marked on top of each measure: *, *p* < 0.05.

### 4.3. Dominance Rating

The dominance ratings for white to coloured light (all three groups) showed decreases with statistically significant differences (*p* = 0.029) and a moderate effect size (Wilcoxon effect size = 0.396). There were no statistically significant differences in Group 1 (white to green–blue light); Group 2 (white to red–green light); and Group 3 (white to red–blue light). Group 1 and Group 2 had a moderate effect size (Wilcoxon effect size = 0.305 and 0.378, respectively) and Group 3 presented a large effect size (Wilcoxon effect size = 0.504). The male group presented statistically significant decrease differences in dominance score (*p* = 0.007) in large effect sizes (Cohen's d = −0.75). The significance and effect size results of the other groupings analysed showed little variation in dominance ratings (Figure 8).

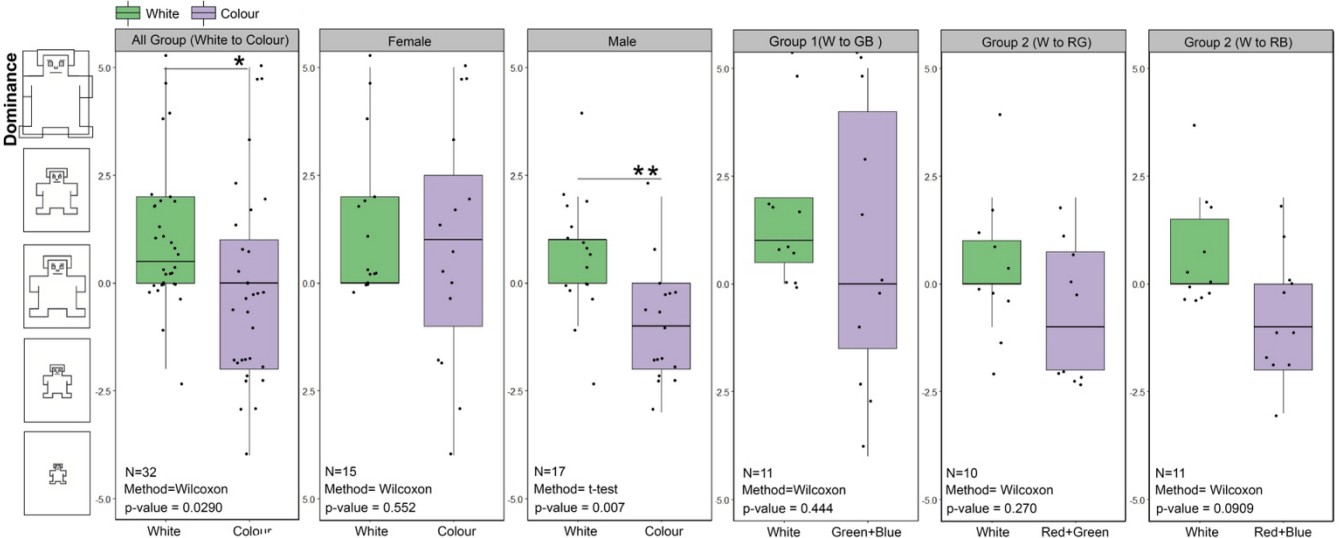

**Figure 8.** Boxplots of dominance ratings from white to coloured light (5 = strong dominance; −5 = weak dominance). W to GB, white to green–blue light; W to RG, white to red–green light; W to RB, white to red–blue light; N, number of people. The statistical significance of repeated t-test or Wilcoxon test is marked on top of each measure: *, *p* < 0.05; **, *p* < 0.01.

### 4.4. Comfort Rating

The comfort ratings for white to coloured light (three groups) showed decreases without statistically significant differences (*p* = 0.132) and a small effect size (Wilcoxon effect size = 0.27). There were no statistically significant differences in Group 1 (white to green–blue light); Group 2 (white to red–green light); and Group 3 (white to red–blue light). Group 1 had a very small effect size (Cohen's d = 0.13) and Group 2 and Group 3 had a small effect size (Cohen's d = 0.44 and 0.26, respectively). The significance and effect size results of the other groupings analysed showed little variation in comfort ratings (Figure 9).

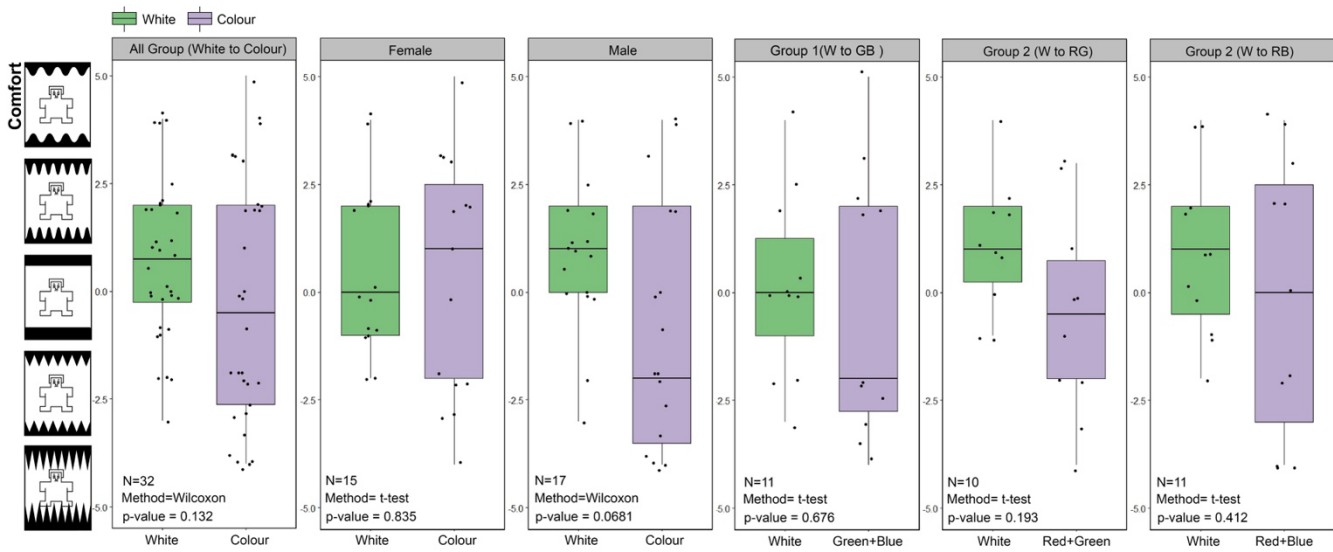

**Figure 9.** Boxplots of comfort ratings from white to coloured light (5 = very comfortable; −5 = very uncomfortable). W to GB, white to green–blue light; W to RG, white to red–green light; W to RB, white to red–blue light; N, number of people. The statistical significance of repeated t-test or Wilcoxon test is marked on top of each measure.

### 4.5. Dimensional Model of Emotion Converted to Basic Emotion

We used Mehrabian and Russell's PAD/VAD model to convert participants' emotion dimensions into basic emotions and analysed the change of mood in a group whenever there was a statistically significant change in either pleasure, arousal, or dominance. A total of 32 participants in three groups in the experiment showed significant differences in both arousal and dominance ratings with respect to white and coloured light. The correspondence between the PAD/VAD model and the emotion dimension of emotions is provided by Russell and Mehrabian [48]. The relationship between the values of the six basic emotions and valence, arousal, and dominance is shown in Table 5 below.

**Table 5.** Values (−1 to 1) for the six basic emotions in terms of emotion dimensions.

|          | Valence | Arousal | Dominance |
|----------|---------|---------|-----------|
| Anger    | −0.43   | 0.67    | 0.34      |
| Joy      | 0.76    | 0.48    | 0.35      |
| Surprise | 0.4     | 0.67    | −0.13     |
| Disgust  | −0.6    | 0.35    | 0.11      |
| Fear     | −0.64   | 0.6     | −0.43     |
| Sadness  | −0.63   | −0.27   | −0.33     |

Through classification with the *k*NN method, six base emotions were classified by the PAD/VAD model. The emotions of the 32 participants were clustered into six groups of basic emotions, and the positioning of each participant's emotion in the PAD model is shown in Figure 10. The percentages of the six emotions (surprise, disgust, joy, fear, anger, and sadness) in white light accounted for 46.88%, 21.88%, 15.63%, 0%, 3.11% and 12.5%; 34.36%, 9.38%, 9.38%, 12.5%,12.5% and 21.38%, respectively, in coloured light. By comparing each emotion, we could obtain a clearer picture of the difference in mood, from white light to coloured light: surprise decreased by 12.52%, disgust by 12.5% and joy by 6.25%, while fear increased by 12.5%, anger by 9.39% and sadness by 9.38%. The overall change in mood was negative, with joy showing the least change and surprise, and disgust and fear having the greatest impact. The effect on mood was more significant for the male subgroup, with the six emotions (surprise, disgust, joy, fear, anger and sadness)

accounting for 37.88%, 11.06%, 25.53%, 0%, 0% and 25.53% in the white light and 30.38%, 5.88%, 5.88%, 25.53%, 5.88% and 26.45% in coloured light, with a 7.50% decrease in surprise from white to coloured light, a 5.18% decrease in disgust, a 19.65% decrease in joy, a 25.53% increase in fear, a 5.88% increase in anger and a 0.92% increase in sadness. The smallest impact was sadness, with fear and joy changing by 20–25%. (Table 6).

**Table 6.** Basic emotions of participants in white light versus colour.

| Category of Emotion | All Groups | | | Male Group | | |
|---|---|---|---|---|---|---|
| | White Light | Coloured Light | Difference | White Light | Coloured Light | Difference |
| Surprise | 46.88% | 34.36% | −12.52% | 37.88% | 30.38% | −7.50% |
| Disgust | 21.88% | 9.38% | −12.5% | 11.06% | 5.88% | −5.18% |
| Joy | 15.63% | 9.38% | −6.25% | 25.53% | 5.88% | −19.65% |
| Fear | 0% | 12.5% | 12.5% | 0% | 25.53% | 25.53% |
| Anger | 3.11% | 12.5% | 9.39% | 0% | 5.88% | 5.88% |
| Sadness | 12.5% | 21.88% | 9.38% | 25.53% | 26.45% | 0.92% |

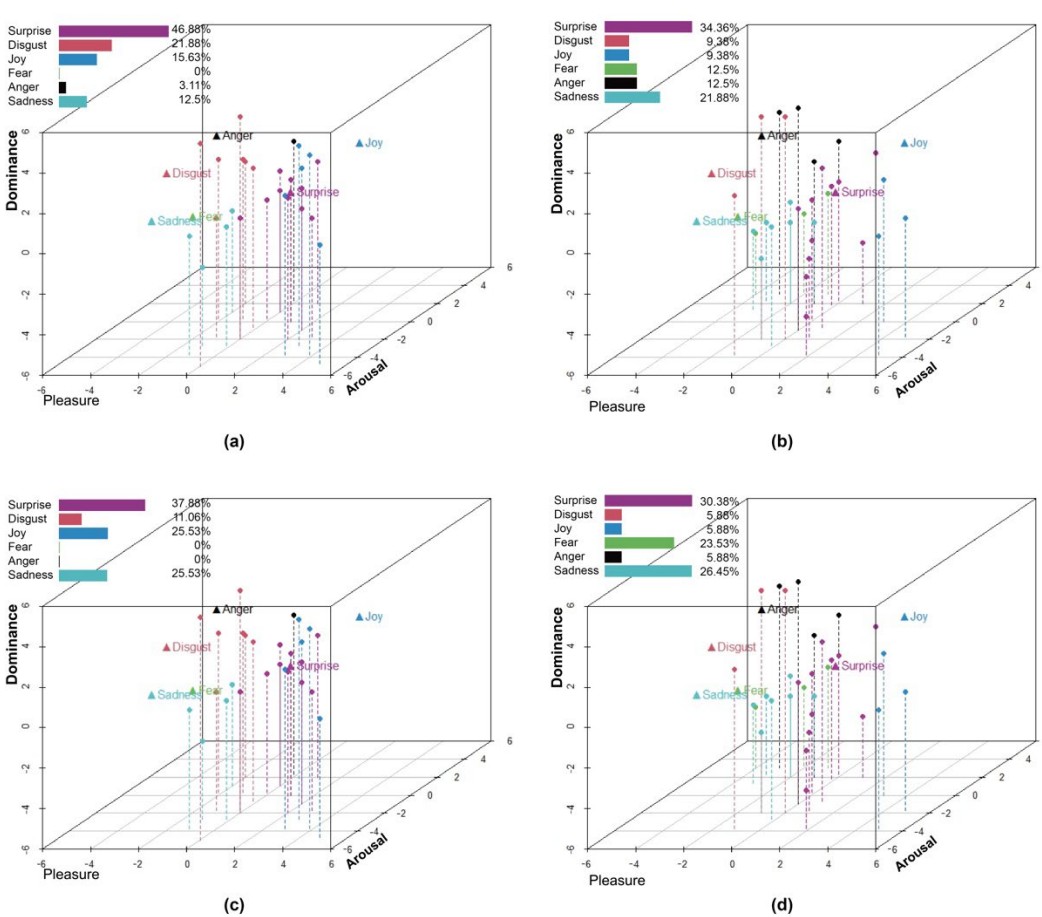

**Figure 10.** Distribution of instances from white to coloured light in the PAD/VAD space visualized according to emotional category. (**a**) Emotion in white light (All group). (**b**) Emotion in colour light (All group). (**c**) Emotion in white light (Male group). (**d**) Emotion in colour light (Male group).

## 5. Discussion

Our waiting room experiments showed that the emotions of the participants were significantly negatively affected by the change from white to coloured light, with joy decreasing and fear, anger and sadness increasing, especially for male groupings, in which

fear and sadness were both substantially elevated. Overall, in terms of comfort, no significant differences between white and coloured light were observed in the results for all sub-groups. This was inconsistent with the hypothesis that the use of coloured ambient light in waiting spaces can alleviate negative emotions.

The increase in negative emotions in the experiment was mainly reflected in the use of coloured lighting, with an increase in arousal values and a decrease in dominance values; meanwhile, no statistically significant change in pleasure rating was obtained during the experiment. The changes in arousal values were consistent with previous findings [35–37]. In the study of Valdez and Mehrabian [49], the saturation of the colour dimension had a strong effect on self-rated arousal, with higher saturation corresponding to higher arousal levels; our data support the conclusion that light colour saturation is positively correlated with arousal values. We also found larger arousal differences in female participants than in males; however, these results did not present statistical significance. Most previous studies used the Circumplex model (Valence–Arousal) analysis [50], thus ignoring the effect of dominance values on emotion, so most findings are limited to changes in alertness and arousal. We classified the data using the PAD/VAD model, converting the PAD/VAD values into six basic emotion categories. In the results, we were able to compare the changes in these basic emotion categories, not just the VA values, such that we could visualise the change in participant emotions with respect to white and coloured light. For example, we presented the results of the experiment as a radar chart, from which non-researchers could very easily understand the changes in each emotion (Figure 11). In this experiment, we did not find that coloured light improved the negative emotions of the participants in the waiting room as we had expected. However, using a dimensional model of emotion to analyse the data and displaying the results in a discrete model of emotion does allow us to express the emotions of people more easily in interior spaces.

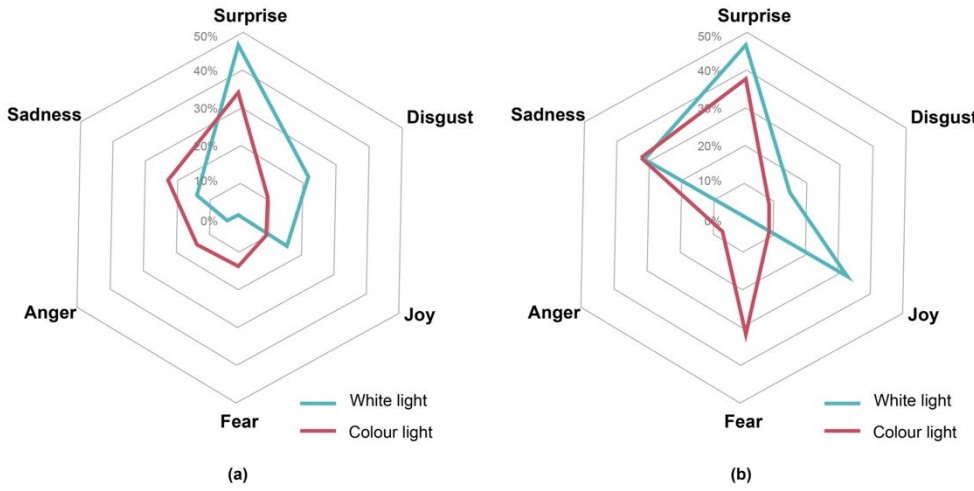

**Figure 11.** Radar plot of participants' emotions in white versus coloured light. (**a**) Participants' emotions in white versus coloured light. (**b**) Male participants' emotions in white versus coloured light.

Although our experiments used realistic scenarios—and, most importantly, white light as a reference value for coloured light changes, proposing the use of a dimensional model of emotion for the analysis and a discrete model of emotion for the results—in contrast to most previous studies, there were still limitations to our experiments. (1) We only considered coloured light versus white light, rather than considering a combination of coloured light plus natural light, or a combination of coloured light plus white light versus white light; in practical use scenarios, indoor coloured light parameters such as CRI and CCT struggle to meet the requirements of EN 12464-1. (2), No significant changes in emotion were found for particular colour combinations of lighting in this experiment, which may be due to the inadequate sample size of the study. Another problem with the

inadequate sample is the precision of the results, as we used clustering to classify the basic emotions, but instead of giving a weighted value to these basic emotions, we simply clustered them. In order to improve the precision of the clustering, the sample size should be increased. (3) To reduce the emotional impact of other factors on participants, we used a windowless room. The effect on participants' moods may differ between the windowless room and the windowed room environment. (4) The experiment was in a non-clinic waiting room setting, where participants were in a healthy emotional state, and, therefore, might not have the same results in a clinic waiting room, so this result applies only to the emotional impact on healthy people. (5) In our experiments, in order to obtain a high-quality white light, our criterion was to choose LED white lights with a CRI > 90; there are not many LED lights on the consumer market with CRI > 90. The LED lights model with CRI > 90 that we were able to obtain in Barcelona was the OSRAM LED superstar plus classic stick bulbs. After choosing the bulb for the white light, we tried to choose a similar bulb for the coloured light to match the white light as closely as possible. However, as uneven lighting in the environment causes eye fatigue [51], LED strips distribute light more evenly than LED bulbs. Therefore, in future research, it will be necessary to consider the use of LED strips as a light source instead of LED bulbs. (6) Although some studies have shown that memory, alertness, attention span, reaction time, learning ability and cognitive ability all perform better under blue light, blue light suppresses melatonin secretion and prolonged blue light use may lead to retinal cell damage [52,53]. It is, therefore, necessary to consider the long-term effects of blue light in coloured ambient light on the user, and not to ignore the negative effects of coloured light beyond emotions.

Several recommendations for future research are given. (1) To meet the indoor lighting requirements of EN 12464-1 [39], consider using a combination of white lights as the primary and coloured lights as secondary luminaires. (2) Increase the sample size of participants; especially with the ageing of the European Union, the number of participants over 60 should be increased appropriately. (3) Previous studies have found inconsistent effects of coloured light on people's moods, and our study is one of the few to obtain negative effects; perhaps a more appropriate psycho-physical methodology should be considered. (4) Use LED strips instead of LED bulbs. (5) Take into account the negative effects of blue light when designing coloured light experiments. (6) Consider adding weighting values when using the *k*NN clustering algorithm.

In conclusion, our study supports that the coloured lights in the waiting room had a systematic effect on the emotional state of the participants. The hue of the coloured light had a significant effect on arousal and dominance, and no significant effect on valence and comfort. Clustering into basic emotions by the PAD model showed that participants experienced negative effects from white to coloured lights. For males in particular, joy in coloured light decreased by 20% and fear increased by 20%. We also suggest that using a dimensional model of emotion to analyse the data and displaying the results in a discrete model of emotion makes it easier to express the emotions of people in interior environments.

**Author Contributions:** Conceptualisation, Z.Z., J.M.F.M. and L.G.M.; methodology, Z.Z. and J.M.F.M.; software, Z.Z.; validation, J.M.F.M. and L.G.M.; formal analysis, Z.Z.; investigation, Z.Z.; resources, Z.Z.; data curation, Z.Z.; writing—original draft preparation, Z.Z.; writing—review and editing, Z.Z., J.M.F.M. and L.G.M.; visualisation, Z.Z.; supervision, J.M.F.M. and L.G.M.; project administration, Z.Z.; All authors have read and agreed to the published version of the manuscript.

**Funding:** This research received no external funding.

**Institutional Review Board Statement:** The study was conducted according to the guidelines of the Declaration of Helsinki and approved by the Ethics Committee of the UPC (21/06/2022).

**Informed Consent Statement:** Informed consent was obtained from all subjects involved in the study.

**Data Availability Statement:** The data presented in this study are openly available in [FigShare] at [10.6084/m9.figshare.20357589].

**Conflicts of Interest:** The authors declare no conflict of interest.

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
