# Peer review of "The Effects of White versus Coloured Light in Waiting Rooms on People’s Emotions"

_buildings, doi:10.3390/buildings12091356_

Round 1

Reviewer 1 Report

This study is very meaningful for the study of the light environment in the waiting room. We all know that light has direct and indirect effects on human mental behavior, hormones, and circadian rhythms. In particular, medical institutions cause a lot of mental and emotional tension and stress to patients. It is a simple and effective method to reduce the stress of waiting for patients through the ambient light that is used every day. However, using lights of different colors to form the light environment of a medical institution makes people feel counterintuitive. As we all know, patients and their families in the waiting room are in a relatively sensitive state of emotions and spirits, and it is impossible to achieve the expected results when using a high-contrast lighting environment.

The following are suggestions for editing the text:

1. Color temperature and correlated color temperature have different meanings, and the author should not use them in confusion.

2. The format of the title in Figure 9 does not match. Please pay attention to the format of the pictures and tables in this article.

3. How can t-test and Wilcoxon test be compared and analyzed together

4. Chapter 1 spends too much space on literature review, and most of the colors are different because of the different colors of the walls, the colors that appear on the screen, and the reactions to people's emotions. It doesn't really fit the research topic.

5. It is a pity that the advantages claimed by the author in "1.4. Current Gaps in Knowledge and the Potential Contribution of the Proposed Approach" were not presented in the final results.

6. Authors are suggested to refer to IEC 62471:2006/CIE S 009:2002 Photobiological safety of lamps and lamp systems, or TECHNICAL REPORTS provided by CIE official website. It will be helpful for your experimental design.

Reviewer 2 Report

Comments about the work are posted in the file.

Reviewer 3 Report

I have analysed this article about the effect of coloured light on People’s emotions. 

Generally speaking, the article is well researched.

There are some minor problems with inadequate expressions that appear here and there like for instance:

“wore vision-corrected eyes”

What is the meaning of this expression? Please clarify

The graphic part of the article seems slightly insufficient. The waiting room of the experiment is not well depicted in plan but later on the authors speak of "architecture". Artificial lighting influences architecture but a windowless waiting room is perhaps an awkward example of "architecture", one feels that other factors like views, vegetation, natural light, furniture or smell and sound might be more influential in the problem that white light or coloured light. 

The authors have developed a correct methodology. However, they do not explain sufficiently the procedures; the sample is small (around 30 people) and we have few details about the provenance of the subjects selected, f.i. do they all come from Barcelona? Are some of them newcomers? A more appropriate psycho-physical methodology should be considered.

I fear that it is premature to conclude that colour saturation is negative for people’s emotions. Dealing with a waiting room, the actual persons who get there have a purpose that the subjects in the experiment, being in perfect health, do not have. In my humble opinion, this experiment cannot be extended much more in its scope and goals unless the setting are altered.

For the rest the article shows no special problems. The references seem concise although adequate.

The results should be checked on the light of what has been exposed before.

Summary of evaluation: This article is promising but it might lack control in certain assumptions and experiments. I have identified some issues in the discussion and methodology. I suggest that this article be accepted for publication after minor revisions.

Author Response

Dear Reviewer #3

All of your questions were answered one by one. However, because we received comments from a new reviewer on 12 August 2022, it will be a week before I  upload the revised manuscript. 

Round 2

Reviewer 1 Report

1.         Figure 1(a)(b) surprised me. It turned out that the author used different sizes of lights to mix light. Can the author explain why this method is used?

2.         There are many hidden LED strips on the market, which can be dimmed and changed in color. The color of mixed light and changing light is very suitable, and it can be used as indirect lighting.

3.         In addition, hospital lighting pays more attention to the uniformity of the light source distribution, and considering the efficiency of the impact on the human eye, light not only affects the psychology, visual physiology, hormones and physiological clocks are all related to the impact of light on the human body.

4.         For example, the spectrum of light sources in Figure 2, short-wavelength blue light is located in the short-wavelength of bad blue light, only green light and red light group, will not affect the circadian rhythm and mood of the human body after long-term exposure, in order to improve the mood of patients in the waiting room, and The impact on the health of medical staff who work in the hospital for a long time is a future research plan that needs to be taken into account by the authors.

5.         Figure 2 Title number ran away.

6.         Figure 3 title ran away.

7.         Figure 8. The picture is repeated.

8.         Figure 9 has one more title legend.

9.         Line 15 of page 14 ran away.

Reviewer 2 Report

All comments of the reviewer were included in the text of the work.

The responses to individual comments are satisfactory.

The authors have taken the trouble to include the relative spectral characteristics of the light sources, which deserves praise.

Remarks

1. In the graphs (Fig. 2.) with the vertical axis it will be better to write "Relative spectral power distribition" or "Relative values"

When describing the horizontal axis, it is useful to insert a space between the words "Wavelegth" and "(nm)".

Please also try to thicken the graduation. Currently, there are only three values ​​at the vertical axis: 380 nm, 580 nm and 780 nm.

2. In table 1 (part ①), there is "Bright-ness of light". Shouldn't it be "illuminance" since the values ​​in the "Experiment Conditions" column are in lx?

3. Figure 2b (page 7.) is not the light curve of the luminaire. Although the drawing looks good visually, it is not correct in terms of content. Figure 2b shows that a significant part of the luminous flux (over 90%) is directed upwards. This would mean that the lampshade does not transmit light or transmits it very little.

In other words, fig 2b shows that it is a luminaire for indirect lighting. Looking at the photos in Fig. 2a, it can be assumed that it is a semi-indirect luminaire.

It is true that carrying out measurements requires specialized equipment.

Due to the inability to carry out measurements and the lack of manufacturer's data, I suggest that you do not use the chart (Fig. 2b).

The text of the work can contain information (one sentence) about the luminaire that it is: semi-indirect.

4. Format (Page 8) "Abbreviations: CRI = color rendering index." It may raise some doubts. In place of the "=" sign, consider putting a hyphen („-“).

After making corrections, recommends the article for publication.
